# Comparison of interproximal delivery and flow characteristics by dentifrice dilution and application of prepared toothpaste delivery technique

**Ryouichi Satou** [1]*, **Atsushi Yamagishi**[1], **Atsushi Takayanagi**[1], **Seitaro Suzuki** [1], **Dowen Birkhed**[2], **Naoki Sugihara**[1]

**1** Department of Epidemiology and Public Health, Tokyo Dental College, Tokyo, Japan, **2** Department of Cariology, University of Gothenburg, Gothenburg, Sweden

* satouryouichi@tdc.ac.jp

**Data Availability Statement:** All relevant data are within the paper and its Supporting Information files.

## Abstract

In home care, the toothpaste technique, which can enhance the caries-preventive effect without changing the amount of dentifrice and fluoride ion concentration, is of great significance. This study aimed to construct a model and experimental system that reproduces the interdental part and to clarify the relationship between the change in dentifrice viscosity due to dilution and washout in the high-risk approximal area of caries. Additionally, the effectiveness of the toothpaste technique and appropriate devices for delivering dentifrice to the interdental area at a low dilution were investigated. Diluted toothpaste samples were prepared (: ×1.00, ×1.25, ×1.50, ×1.75, ×2.00, ×3.00, and ×4.00). An acrylic interproximal model was created for this experiment. The flow characteristics and viscosity by dentifrice dilution were measured. In the case of low dilution of 57% (1.75×) or more, it was shown that the dentifrice in the high-risk area may be washed out early because of the decrease in viscosity, and the caries-preventive effect may be reduced. It was also suggested that to keep the dentifrice in the interdental area for 120 s at the end of brushing, a dilution must be devised to a concentration of at least 50% (2.00×). The prepared toothpaste delivery (PTD) method of delivering dentifrice to the interdental area while maintaining it at a low dilution is an effective toothpaste technique in terms of dentifrice dilution and viscosity. The use of finger brushes in the PTD method could increase the efficiency of dentifrice delivery.

## Introduction

The prevalence of caries in children is decreasing because of the widespread of fluoride-containing dentifrices [1]. It has been reported that the caries prevention of fluoride-containing dentifrice is as high as 30–40% [2]. Additionally, the application of fluoride-containing dentifrice is cost-effective by a result of the analysis with DMFT index as the outcome [2]. Therefore, home care is anticipated to be important not only in developed countries but also in developing countries, where the prevalence of dental caries is still high [1–3]. However, there

**Funding:** The authors received no specific funding for this work.

**Competing interests:** The authors have declared that no competing interests exist.

are few reports on toothpaste techniques compared to the examination of dentifrice fluoride concentration and brushing methods.

The toothpaste technique can enhance the caries-preventive effect by efficiently applying the active ingredients of dentifrice to high-risk areas [4]. There is a positive correlation between the fluoride ion concentration and caries suppression rate of fluoride-containing dentifrices [2, 5]. However, in children with high fluoride ion absorption, increasing the fluoride ion concentration of the dentifrice may increase the risk of fluoride exposure [3]. Therefore, there is a need to develop a toothpaste technique that can enhance the caries preventive effect and the active ingredients of dentifrice without changing the amount of dentifrice and the fluoride ion concentration.

Sjögren and Birkhed have developed a toothpaste technique called the modified fluoride toothpaste technique (MFTT), in which a slurry rinse is performed using fluoride toothpaste after brushing [4]. The MFTT has been reported to enhance the effectiveness of fluoride toothpaste and reduce adjacent interdental caries in preschool children by an average of 26% [6–8]. Additionally, Al Mulla et al. reported using a similar method, 2+2+2+2 [9]. These toothpaste techniques enhance the caries effect of dentifrices by maintaining high levels of fluoride in the oral cavity for extended periods.

The interdental area and pit fissure are high-risk areas for caries, and the tips of toothbrushes are difficult to reach [10]. Therefore, the effects of fluoride-containing dentifrice and mouthwash are important for caries prevention in these areas. To enhance the effect, the drug must stay in the high-risk area for as long as possible [11]. This study aimed to construct a model and experimental system that reproduces the interdental part and to clarify the relationship between the change in dentifrice viscosity due to dilution and washout at the high-risk part of caries. Additionally, the effectiveness of the toothpaste technique and appropriate devices for delivering dentifrice to the interdental area at a low dilution were investigated.

## Materials and methods

### Preparation of toothpaste and dental instrument

The toothpaste used in this study was Syumitect Complete ONE (GlaxoSmithKline plc., London, England) with 1500 ppm F in the form of sodium fluoride, which is commonly available in Japan. The toothpaste was diluted stepwise with ion-exchanged water to study its viscosity and decline rates. Seven samples were prepared with the following concentrations: 1.00, 1.25, 1.50, 1.75, 2.00, 3.00, and 4.00. In this experiment, a commercially available general GUM dental brush #211 toothbrush (three-row compact head, normal, GUM Corp., Tokyo, Japan) and a finger brush (Deep Clean, M size, KAO Corp., Tokyo, Japan) were used. The finger brush is made of silicone and shaped like a finger cap.

### Operators

Five right-handed dentists of the Tokyo Dental College, familiar with the use of a toothbrush and finger brush, performed the experiments. The operators had sufficient experience in the prepared toothpaste delivery (PTD) technique. The brushing methods were calibrated between the operators.

### Measuring the viscosity of toothpaste

A sine-wave vibro (tuning fork-type vibration) viscometer (SV-10, A&D Corp., Tokyo, Japan) was used to measure the viscosity of the toothpaste. The measurements were conducted at 25˚C according to the manufacturer's instructions.

## Production of acrylic interproximal model and black jaw model

An acrylic interproximal model and black jaw model were created for this experiment. An acrylic interproximal model was constructed to observe the flow characteristics of the dentifrice in this area. This model consisted of two acrylic plates (10 mm × 10 mm) and a 1-mm spacer. The space between the two acrylic plates reflected the human interproximal area and was adjusted such that the gap is 516 μm at a position 4.5 mm from the bottom (S1 Fig). The gaps in this model can be filled using dentifrices. The model was glued to an acrylic stick, and the orientation was changed by rotating the acrylic stick. One side of the acrylic stick was colored red to emphasize the contrast with white toothpaste.

The black jaw model was created to measure the difference in dentifrice delivery depending on the device used in the PTD technique [12]. Dental jaw model for conservation and restoration training (D18FE-500E, NISSIN, Tokyo, Japan) was painted with surface primers for plastic (Surface primer No. 26, TAMIYA, Shizuoka, Japan) and black spray (TAMIYA color No. 06 Black, TAMIYA, Shizuoka, Japan). The color of the model was black to emphasize the contrast with the white toothpaste.

## Comparison of flow characteristics by dentifrice dilution

Fifty milligrams of toothpaste was filled into the gap of the acrylic interproximal model. A 5-L beaker was charged with 4-L of ion exchange water, and a constant water flow was generated at a stirrer rotation of 1500 rpm. The acrylic interproximal model was held in water at a depth of 5 cm below and parallel to the water surface, and the residual amount of toothpaste was measured every 15 s for a maximum of 180 s (S1 Fig). The model was rotated 180° with respect to the water flow every 15 s so that the amount of dentifrice on the left and right sides was not biased. Photographs were taken using a digital camera (ILCE-7M3, Sony, Tokyo, Japan) with microlens (AI Micro-Nikkor 55 mm f/2.8S, Nikon, Tokyo, Japan) horizontally 10 cm above the acrylic interproximal model. Images were taken to analyze the amount of dentifrice remaining in the gap. The residual area ratio of the dentifrice (%) was measured using image analysis software (Image J, version 1.52) (S2 Fig).

## Measuring the difference in dentifrice delivery depending on the device used in the PTD

In this experiment, PTD was performed on a black jaw model using two types of devices: a toothbrush and a finger brush, and the dentifrice delivery properties in the interproximal area were compared. The procedure of the PTD technique is as follows: (1) 1.0 g of toothpaste is placed on the brush head, (2) the undiluted toothpaste is applied to all teeth, starting from the buccal interproximal space of the molar region, (3) the teeth are brushed for 2 min (S3 Fig). At the end of the PTD, images were taken using a digital camera (ILCE-7M3, Sony, Tokyo, Japan) with a microlens (AI Micro-Nikkor 55 mm f/2.8S, Nikon, Tokyo, Japan) 15 cm above the model. All images were captured with a linear polarizing filter attached to the lens and imaging light to avoid reflections of the acrylic model surface. All interproximal areas of the model were evaluated with the "PTD score" (described below), and the average score for each device was calculated.

## Evaluation of dentifrice delivery by PTD score

We created a PTD score to assess the deliverability of the dentifrices in the interproximal area. This score covers only the interproximal area and examines seven points on the right side (R12, R23, R34, R45, R56, R67, and R78), one place in the middle (C11), and seven places on

the left side (L12, L23, L34, L45, L56, L67, L78), for a total of 15 places. There were three scores: 0, 0.5, and 1. A score of 1 indicated that the lingual interdental space was filled with dentifrice. A score of 0.5 indicated half of the space and a score of 0 indicated that the space is completely free of dentifrice (S3 Fig). The total score at the examination site of the mandibular model was calculated.

## Statistical analysis

In the experiment of flow characteristics by dentifrice dilution, the statistical analysis among the test groups was performed by one-way analysis of variance, and differences were considered significant at $p < 0.05$. Data for each group are presented as the mean ± SD of three replicates (n = 3). The Bonferroni test was used for post-hoc comparisons when significance was determined using analysis of variance ($p < 0.05$). PTD scores are presented as the mean ± SD of three replicates per brushing device (n = 3). A significant difference test was performed using the paired t-test ($p<0.05$).

## Results

### Viscosity of dentifrice according to the dilution ratio

Fig 1 shows the relationship between the dilution ratio of dentifrice and viscosity (Fig 1). For a dentifrice with a concentration of 100% (1.00×, no dilution), the viscosity of the dentifrice showed the highest value of 4810 mPa·s (Fig 1A). When the dentifrice was diluted to a concentration of 80% (1.25×), the viscosity decreased sharply to 995 mPa·s, approximately one-fifth of 100% (1.00×). When diluted to a concentration of 67% (1.50×), the viscosity decreased to 396 mPa·s, which was less than half of that at 1.25×. The viscosity at a concentration of 57% (1.75×) was 174 mPa·s, 50% (2.00×) was 98.5 mPa·s, and 33% (3.00×) was 30.2 mPa·s. Viscosity tended to decrease as the dilution ratio of dentifrice increased (Fig 1A). At 25% (4.00×), which is the diluted concentration of dentifrice assuming the oral cavity after brushing for 2–3 min, the viscosity was the lowest at 13.1 mPa·s. Focusing on the change in viscosity, the decrease in viscosity from 100% (1.00×) to 80% (1.25×) was the most remarkable, and after 67% (1.50×), the viscosity gradually decreased. A scatter plot is shown, in which the concentration (%) is on the horizontal axis and the logarithmic value of the viscosity is on the vertical axis to describe the correlation between the dilution ratio of the dentifrice and the viscosity (Fig 1B). In this experiment, we found a strong positive correlation between dentifrice concentration and viscosity (r = 0.999).

### Relationship between dentifrice dilution and residual area ratio of dentifrice in the interdental model

Fig 2 shows the time of exposure to the water flow and the remaining area of the dentifrice in the interdental model. This model reproduced the interdental part by forming a gap between the two acrylic plates. In this experiment, it was possible to compare the extent to which the dentifrice filled the gap at each dilution ratio under the same conditions and at the same time (S1 and S2 Figs). Fig 3 shows the residual area ratio of the dentifrice at 60, 120, and 180 s for each dilution ratio (Fig 3). At concentrations of 100% (1.00×, undiluted) and 80% (1.25×), almost all of the dentifrice remained after exposure to a constant stream of water for 180 s (Fig 2). At a concentration of 100% (1.00×), the residual area ratio was 100% at all of 60, 120, and 180 s (Fig 3). At a concentration of 80% (1.25×), there was little change in the residual area ratio, which was 98.9 ± 0.8% in 60 s and 97.2 ± 1.2% in 120 s and 180 s (Fig 3). At a concentration of 67% (1.50×), dentifrice began to be lost from the margin (1 mm wide side) at 30–60 s (Fig 2). The measured values of 67% (1.50×) were 93.6 ± 1.3% at 60 s, 79.2 ± 17.1% at 120 s and

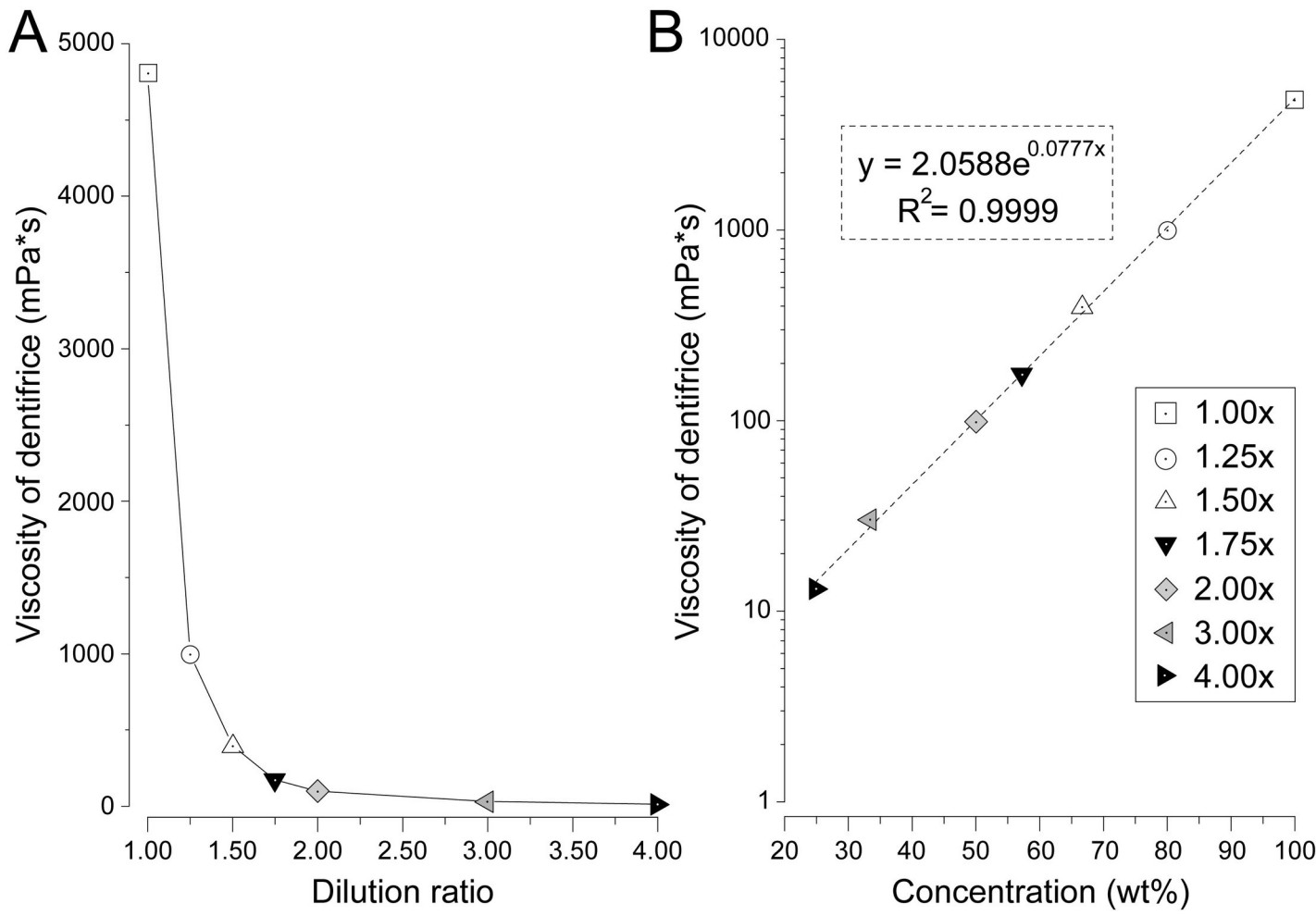

**Fig 1. Effect of viscosity according to dilution ratio of toothpaste.** (A) Relationship between the dilution ratio of the dentifrice and viscosity. (B) Scatter plot with concentration (wt%) on the horizontal axis and the logarithmic value of the viscosity (mPa*s) on the vertical axis to examine the correlation between the dilution ratio of the dentifrice and viscosity. The dotted straight line shows the approximation and R shows the correlation coefficient.

72.3 ± 17.5% at 180 s. After 60 s, a gradual decrease in the dentifrice was observed until the end of 180 s (Fig 3). At a concentration of 57% (1.75×), the dentifrice was significantly washed away after 60 s. The residual area ratio of 57% (1.75×) was 81.9 ± 1.3% in 60 s, decreasing to 51.2 ± 3.8% in 120 s and 41.2 ± 2.6% in 180 s (Figs 2 and 3). The concentration of 50% (2.00×) had already decreased to 43.9 ± 1.8% at 60 s, 27.8 ± 2.2% at 120 s, and 24.0 ± 4.5% at 180 s (Figs 2 and 3). The amount of dentifrice washed out at concentrations of 33% (3.00×) and 25% (4.00×) was remarkable in a short time, and most of the dentifrice did not remain in approximately 30 s after the start (Fig 2). The concentration of 33% (3.00×) was 8.0 ± 2.3% at 60 s, 5.8 ± 1.2% at 120 s, and 3.9 ± 0.9% at 180 s, which were less than 10% at each time (Fig 3). At a concentration of 25% (4.00×), only 1.3 ± 1.6% was measurable in 60 s, and no dentifrice remained after 60 s (Figs 2 and 3).

Fig 4 shows the time transition of the residual area ratio of the dentifrice for each dilution ratio. At concentrations of 100% (1.00×) and 80% (1.25×), there was no significant change in the residual area ratio, and the high residual rate was maintained even after 180 s. When the dilution ratio was 57% (1.75×) or higher, the dentifrice flowed out from 30 s after the start, and the residual ratio decreased in a short time as the dilution ratio increased (Fig 4).

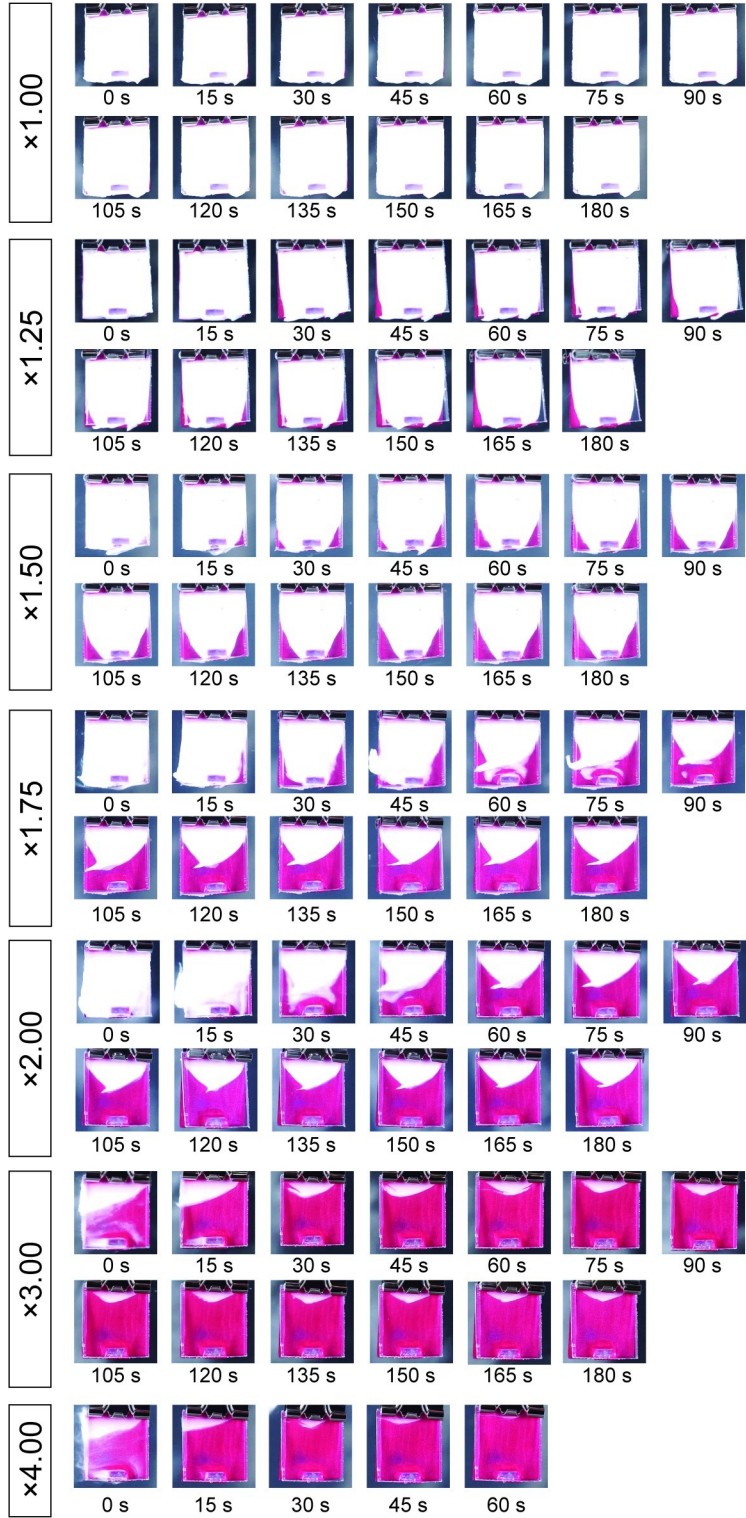

**Fig 2. Flow comparison for each dentifrice dilution rate by interdental model.** The time of exposure to water flow and the remaining area of the dentifrice in the interdental model are shown. This model reproduced the interdental part by forming a gap between the two acrylic plates. The white part indicates the dentifrice filled in the gap, and the red part indicates that the dentifrice has been washed out. Images are shown every 15 s for up to 180 s.

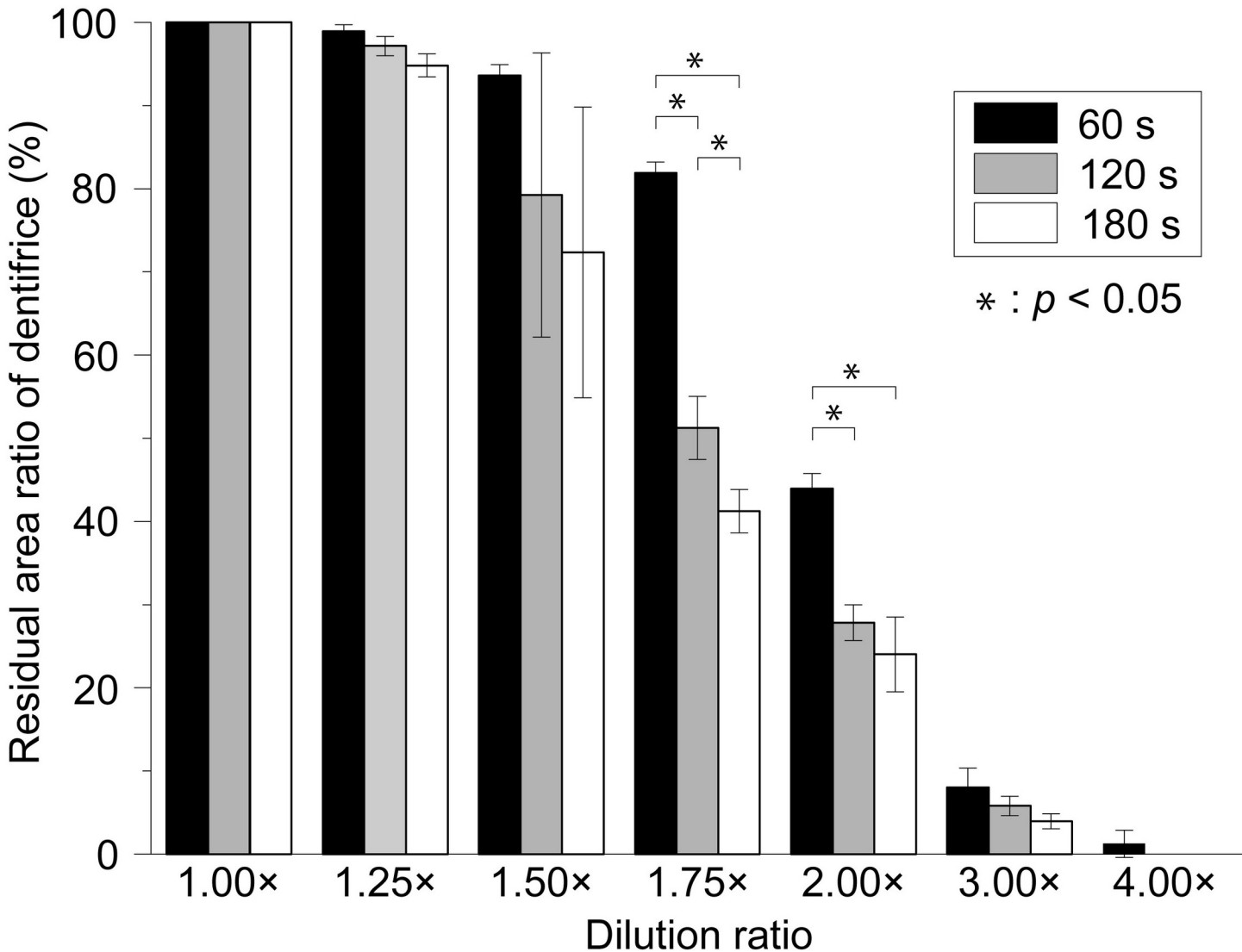

**Fig 3. Dentifrice residual area ratio every 60 s at each dilution ratio.** Fig 3 shows the residual area ratio of the dentifrice (%) at 60, 120, and 180 s for each dilution ratio. Data for each group are presented as the mean ± standard deviation (SD) of three replicates (n = 3). Statistical analysis among the test groups is performed using the analysis of variance (ANOVA) and the Bonferroni test (p < 0.05).

Concentrations of 33% (3.00×) and 25% (4.00×) showed a particularly strong downward trend. At a concentration of 33% (3.00×), the residual area ratio decreased sharply to 64.9 ± 10.3% immediately after the start and 11.6 ± 1.5% in 30 s. The concentration of 25% (4.00×) decreased to 16.4 ± 7.8% immediately after the start, and 2.7 ± 2.6% in 30 s, indicating that most of the dentifrice was flowing (Fig 4). The times required for the residual area ratio to fall below 50% were 135 s at 57% (1.75×), 60 s at 50% (2.00×), 15 s at 33% (3.00×), and 0 s at 25% (4.00×). The three groups with concentrations of 100% (1.00×), 80% (1.25×), and 67% (1.50×) did not fall below 50% in the residual area ratio even after 180 s (Fig 4).

### Measurement of PTD score by the device used during brushing

Fig 5 shows the results of the PTD method on a black jaw model using a toothbrush and finger brush. The procedures of the PTD method and the calculation method for the score are shown

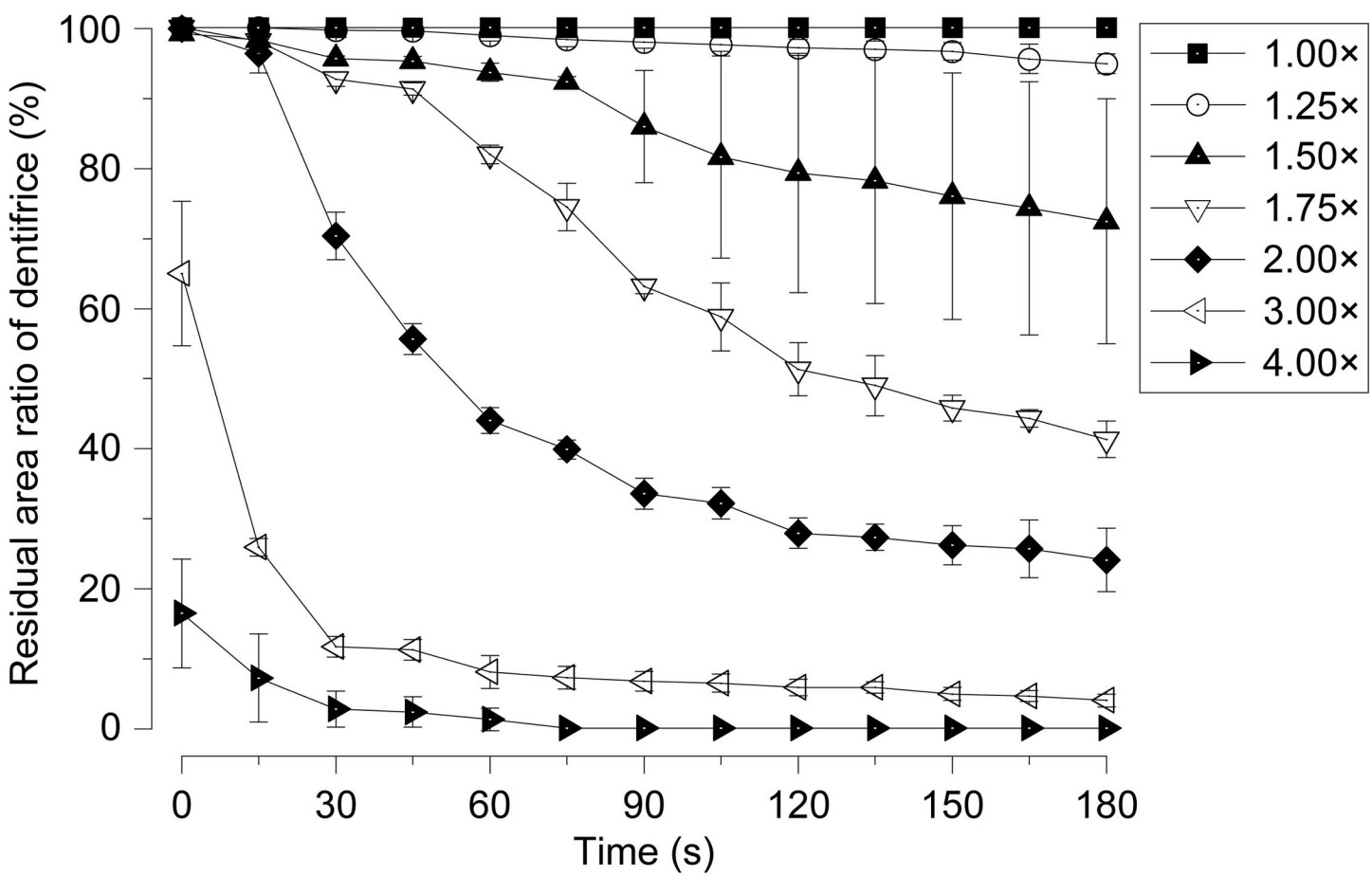

**Fig 4. Time transition of the residual area ratio of the dentifrice for each dilution ratio.** The black square indicates dentifrice with a concentration of 100% (1.00×). White circles: 80% (1.25×), black triangles: 67% (1.50×), white inverted triangles: 57% (1.75×), black diamonds: 50% (2.00×), white left-pointing triangle: 33% (3.00×), black right-pointing triangle: 25% (4.00×). Data for each group are presented as the mean ± standard deviation (SD) of three replicates (n = 3).

in S3 Fig. The average PTD score was 4.67 ± 1.03 when using a toothbrush (Fig 5A). In contrast, the score when using the finger brush was 9.67 ± 0.62, which was significantly higher (p < 0.05). The figure shows the average PTD score ratio (%) calculated by analyzing the images of each interdental area in the model (Fig 5B). The average PTD score ratio was 42.4 ± 9.3% for the toothbrush, which was significantly lower than that for the finger brush (87.8 ± 5.7%) (Fig 5B). The use of a finger brush was able to deliver the dentifrice to approximately 90% of the interdental areas in the jaw model.

## Discussion

### Dilution ratio of dentifrice and persistence in high-risk areas of caries

In this study, it was clarified that the viscosity of the dentifrice decreased sharply due to dilution, and the flow characteristics increased (Figs 1–4). The correlation between dilution factor and viscosity is consistent with the results of our previous studies [12]. Dentifrice is diluted with saliva in the oral cavity and water in bristle on the toothbrush [13]. The longer the brushing time, the greater the diluting effect of saliva because of water. It has been reported that when 1 g of dentifrice is used when brushing the oral cavity, it is diluted approximately 4.00× in 1 min and approximately 5.00× in 2 min [13]. The diluted dentifrice solution can be

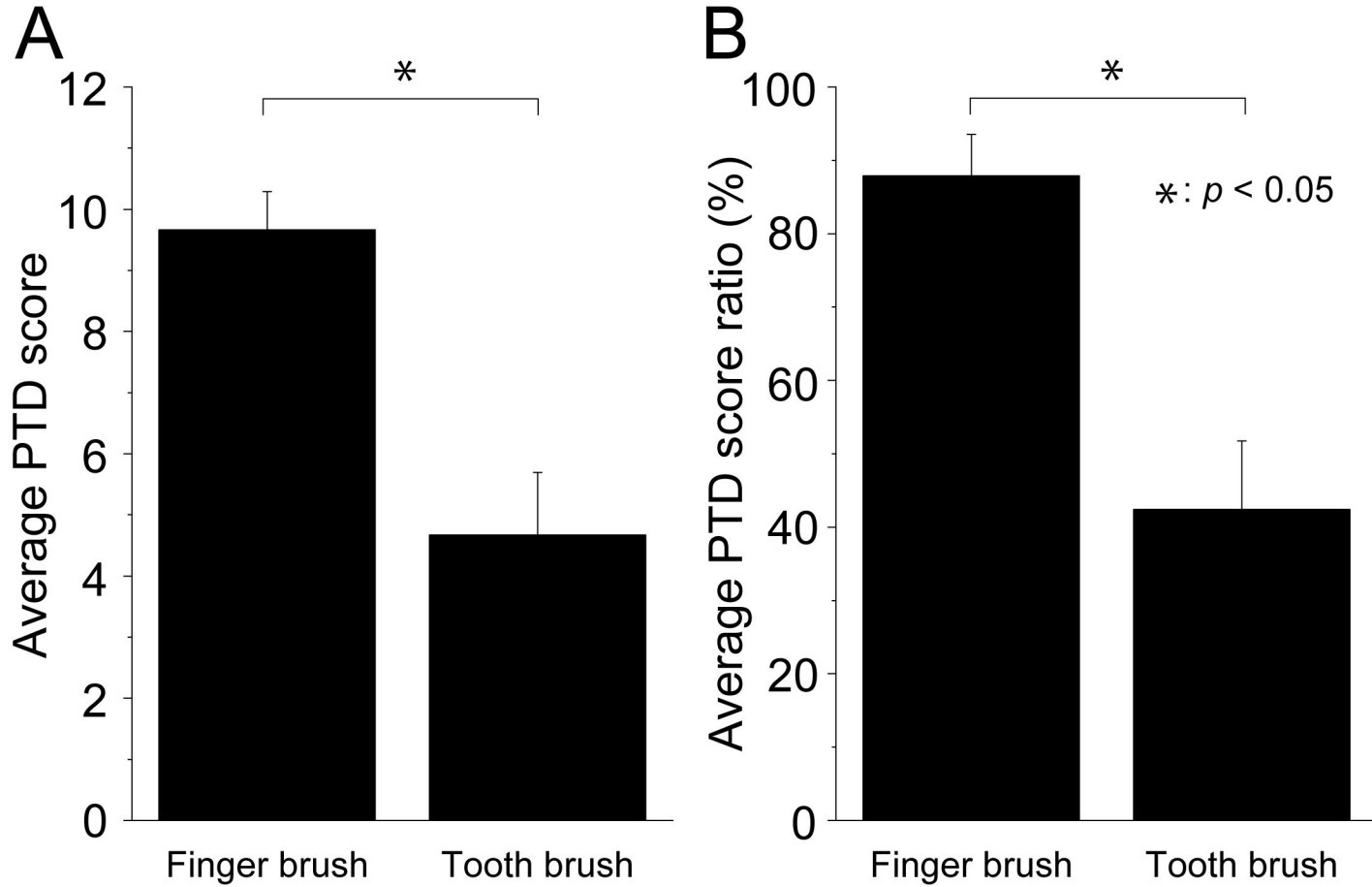

**Fig 5. Comparison of PTD scores by the device used during brushing.** Fig 5 shows the results of the PTD method on a black jaw model using a toothbrush and finger brush. The procedures of the PTD and calculation methods for the score are shown in S3 Fig. PTD scores are presented as the mean ± standard deviation (SD) of three replicates per brushing device (n = 3). A significant difference test is performed using the paired t-test. (p<0.05).

expected to have a chemical effect by delivering it to high-risk caries areas such as interdental areas and pit fissures. However, if the viscosity of the dentifrice is low, it will be washed out immediately [12]. According to the results of this study, when the concentration of dentifrice was 57% (1.75×) or higher, the dentifrice was found to flow out from 30 s after the start. It was also clarified that the higher the dilution ratio, the shorter the residual rate in a short time (Fig 4). This result suggests that the dentifrice in the high-risk area of caries is washed out early in the case of a low dilution of 57% (1.75×) or more. In previous studies, the amount of fluoride ion uptake into the tooth surface was high at concentrations of 100% (1.00×) and 80% (1.25×). However, it has been reported that the dilution ratio was significantly reduced at a concentration of 57% (1.75×) or higher [12]. It was also shown that the uptake of fluoride ions on the tooth surface, when diluted to 5.00×, was reduced to half that of the undiluted solution and that there was a negative correlation between the dilution of the dentifrice and the uptake of fluoride ions. This suggests that the stagnation and caries-preventive effect of the dentifrice decreases at a dilution ratio of 57% (1.75×) or higher. Additionally, the time required for the residual area ratio to fall below 50% was 135 s for a concentration of 57% (1.75×) and shorter than 60 s for 50% (2.00×) or higher (Fig 4). The brushing time recommended by the American Dental Association for the general public is 120 s, and 180 s even in areas that recommend longer times such as Japan and South Korea [13]. From the results of this study, to keep the

dentifrice in the interdental area until 120 s after the end of brushing, it is necessary to devise a dilution to a concentration of at least 50% (2.00×). As this is a model experiment of constant water flow, it is not possible to consider water flow from multiple directions, such as mechanical stimulation by the tip of the brush and gargling. Therefore, the oral cavity is considered to be a harsher environment than this condition. We plan to use our model evaluation system to change the type and devices of dentifrice.

### Selection and clinical application of devices suitable for the PTD method

We have developed and reported a new toothpaste technique aimed at delivering dentifrice to high-risk caries sites with low dilution [12]. The technique, named the PTD method, has an application phase in which the dentifrice is delivered to the interdental area using a toothbrush or finger before the start of brushing (S3 Fig). The application phase of PTD allows the dentifrice to be delivered undiluted to the high-risk caries site and to maintain high viscosity and concentration during brushing. In this study, we compared toothbrushes and finger brushes to determine suitable devices for the application phase (Fig 5). It was suggested that flexible protrusions and elastic silicone finger brushes increased the amount of dentifrice delivered to the interdental area and efficiently filled the voids.

Yamagishi et al. reported that it is necessary to allow fluoride at a concentration of 300 ppm or more to act for 2 min or more to effectively incorporate fluoride ions into the tooth surface [14]. To achieve the conditions using a dentifrice with a fluoride ion concentration of 1000 ppm, it is desirable to use 1.0 g or more [13]. The MFTT recommends rinsing with a small amount of water to deliver fluoride to the interdental area in the form of slurry [4]. In contrast, because the PTD method does not require water volume restriction, the PTD method has a relatively high degree of freedom and is considered to be a method that can be easily incorporated into daily life. The difficulty of the procedure is low because it uses fingers and can be applied not only to adults but also to young children and elderly individuals.

## Conclusion

In the case of low dilution of 57% (1.75×) or more, it was demonstrated that the dentifrice in the high-risk approximal area may be washed out early because of the decrease in viscosity, and the caries-preventive effect may be reduced. To keep the dentifrice in the interdental area for 120 s at the end of brushing, a dilution must be devised to a concentration of at least 50% (2.00×). The PTD method of delivering dentifrice to the interdental area while maintaining it at a low dilution is an effective toothpaste technique in terms of dentifrice dilution and viscosity. We found that the use of finger brushes in PTD could increase the efficiency of dentifrice delivery.

## Supporting information

**S1 Fig. Acrylic interdental model design.** An interdental model is created using two acrylic plates with a 1-mm spacer on one side. The width of the gap changed depending on the distance to the spacer. The gap is approximately 516 μm wide, which is the same as the intraoral tooth spacing at the central 4.5 mm point. The space between the acrylic plates is filled with a dentifrice, and the flow of the dentifrice in a constant stream of water is observed. The model is rotated 180˚ with respect to the water flow every 15 s; therefore, the amount of dentifrice on the left and right sides is not biased. One side of the acrylic plate is painted red; therefore, the outflow of the dentifrice can be easily observed during the image analysis.
(TIF)

**S2 Fig. Calculation method of the residual area ratio of dentifrice by image analysis.** Photographs are captured by a digital camera horizontally 10 cm above the acrylic interproximal model. Images are taken to analyze the amount of dentifrice remaining in the gap. The residual area ratio of the dentifrice (%) is measured using image analysis software (ImageJ, version 1.52). Raw images are converted to eight bits and the white dentifrice portion is measured. The measurement range is 9 mm × 10 mm, excluding the spacer. The signal strength for each pixel is plotted, and the residual area ratio of the dentifrice (%) is calculated.
(TIF)

**S3 Fig. Prepared toothpaste delivery (PTD) technique procedure and scoring method.** (A) The procedure of the PTD technique is (1) 1.0 g of toothpaste is placed on the brush head, (2) the undiluted toothpaste is applied to all the teeth, starting from the buccal interproximal space of the molar region, (3) the teeth are brushed for 2 min. (B) The PTD score is used to assess the deliverability of the dentifrices in the interproximal area. This score covers only the interproximal area and examines seven points on the right side (R12, R23, R34, R45, R56, R67, and R78), one place in the middle (C11), and seven places on the left side (L12, L23, L34, L45, L56, L67, L78), for a total of 15 places. There are three scores: 0, 0.5, and 1. A score of 1 indicates that the lingual interdental space is filled with a dentifrice. A score of 0.5 indicates half of the space and a score of 0 indicates that the space is completely free of dentifrice.
(TIF)

## Acknowledgments

We sincerely thank the Oral Health Science Center, Tokyo Dental College, for allowing us to use their facilities.

## Author Contributions

**Conceptualization:** Ryouichi Satou, Atsushi Takayanagi.

**Data curation:** Ryouichi Satou, Atsushi Yamagishi, Seitaro Suzuki.

**Formal analysis:** Ryouichi Satou, Atsushi Yamagishi.

**Investigation:** Ryouichi Satou, Naoki Sugihara.

**Methodology:** Ryouichi Satou, Atsushi Yamagishi, Atsushi Takayanagi, Seitaro Suzuki.

**Project administration:** Ryouichi Satou, Dowen Birkhed, Naoki Sugihara.

**Software:** Seitaro Suzuki.

**Supervision:** Ryouichi Satou, Atsushi Takayanagi, Dowen Birkhed, Naoki Sugihara.

**Validation:** Ryouichi Satou, Atsushi Yamagishi, Dowen Birkhed, Naoki Sugihara.

**Visualization:** Ryouichi Satou, Atsushi Yamagishi.

**Writing – original draft:** Ryouichi Satou.

**Writing – review & editing:** Ryouichi Satou.

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
