## [Decision Letter · Decision Letter 0]

24 Aug 2022

PONE-D-22-20304Comparison of interproximal delivery and flow characteristics by dentifrice dilution and application of prepared toothpaste delivery techniquePLOS ONE

Dear Dr. Ryouichi Satou,

Thank you for submitting your manuscript to PLOS ONE. After careful consideration, we feel that it has merit but does not fully meet PLOS ONE’s publication criteria as it currently stands. Therefore, we invite you to submit a revised version of the manuscript that addresses the points raised during the review process.

ACADEMIC EDITOR: It is a well written manuscript, however, I would suggest to answer the questions as mentioned by the reviewers.

We look forward to receiving your revised manuscript.

Kind regards,

Tanay Chaubal

Academic Editor

PLOS ONE

Journal Requirements:

2. Please amend your list of authors on the manuscript to ensure that each author is linked to an affiliation. Authors’ affiliations should reflect the institution where the work was done (if authors moved subsequently, you can also list the new affiliation stating “current affiliation:….” as necessary).

Reviewers' comments:

Reviewer's Responses to Questions

**Comments to the Author**

1. Is the manuscript technically sound, and do the data support the conclusions?

Reviewer #1: Yes

Reviewer #2: Yes

2. Has the statistical analysis been performed appropriately and rigorously? 

Reviewer #1: Yes

Reviewer #2: Yes

3. Have the authors made all data underlying the findings in their manuscript fully available?

Reviewer #1: Yes

Reviewer #2: Yes

4. Is the manuscript presented in an intelligible fashion and written in standard English?

Reviewer #1: Yes

Reviewer #2: Yes

5. Review Comments to the Author

Reviewer #1: I congratulate the authors on a well written manuscript.The material and methods section is well described.The discussion brings out well the highlights of the study.The conclusions are in line with the objectives of the study.All in all a well written manuscript.

Reviewer #2: LINE 23 - HOW IS THE FLUORIDE TOOTHPASTE COST EFFECTIVE ANY STATSISTICS?

LINE 230 - ANY STANDARDISATION OF TIME OF BRUSHING IN ALL THE SUBJECTS?

ANY STANDARDISATION OF INTERPROXIMAL DISTANCE OR EMBRASSURE SPACE FOR THE STUDY?

6. PLOS authors have the option to publish the peer review history of their article (what does this mean?). If published, this will include your full peer review and any attached files.

Reviewer #1: No

Reviewer #2: No

---

## [Author Response · Author response to Decision Letter 0]

25 Aug 2022

Response to Reviewers

Reviewer #1:

 I congratulate the authors on a well written manuscript.The material and methods section is well described.The discussion brings out well the highlights of the study.The conclusions are in line with the objectives of the study.All in all a well written manuscript.

> We strongly appreciate the reviewer's comment. We are thankful for the time and energy you expended. We will work hard to make the paper better with this revision.

Reviewer #2: 

>Thank you very much for providing important comments. We are thankful for the time and energy you expended. Our responses to the referees’ comments are as follow:

1. LINE 23 - HOW IS THE FLUORIDE TOOTHPASTE COST EFFECTIVE ANY STATSISTICS?

> We appreciate the reviewer's comment on this point. In accordance with the reviewer's comment, we have changed this and added references [2] to following sentence:

Page 5, Line 42-43

Additionally, the application of fluoride-containing dentifrice is cost-effective by a result of the analysis with DMFT index as the outcome [2].

2. LINE 230 - ANY STANDARDISATION OF TIME OF BRUSHING IN ALL THE SUBJECTS?

>Thank you for providing comments. In this experiment, the brushing time is set to 2 minutes. The operators, dentists, trained and calibrated their methods prior to the experiment. The rationale for the 2-minute period is that clinical guidelines in the U.S. and other countries require 2 minutes for brushing instruction.

Page 7, Line 81-82

The operators had sufficient experience in the prepared toothpaste delivery (PTD) technique. The brushing methods were calibrated between the operators.

3. ANY STANDARDISATION OF INTERPROXIMAL DISTANCE OR EMBRASSURE SPACE FOR THE STUDY?

> In our previous study, we conducted experiments using a model with uniform INTERPROXIMAL DISTANCE and published a paper (Ref.12). In this study, we created a black model to clarify the behavior of toothpaste using a model that reflects the arch and tooth morphology in the oral cavity.

12. Satou R, Suzuki S, Takayanagi A, Yamagishi A, Sugihara N. Modified toothpaste application using prepared toothpaste delivering technique increases interproximal fluoride toothpaste delivery. Clin Exp Dent Res. 2020;6(2):188–96.

---

## [Decision Letter · Decision Letter 1]

4 Oct 2022

Comparison of interproximal delivery and flow characteristics by dentifrice dilution and application of prepared toothpaste delivery technique

PONE-D-22-20304R1

Dear Dr. Ryouichi Satou,

We’re pleased to inform you that your manuscript has been judged scientifically suitable for publication and will be formally accepted for publication once it meets all outstanding technical requirements.

Kind regards,

Tanay Chaubal

Academic Editor

PLOS ONE

Additional Editor Comments (optional):

Reviewers' comments:

Reviewer's Responses to Questions

**Comments to the Author**

1. If the authors have adequately addressed your comments raised in a previous round of review and you feel that this manuscript is now acceptable for publication, you may indicate that here to bypass the “Comments to the Author” section, enter your conflict of interest statement in the “Confidential to Editor” section, and submit your "Accept" recommendation.

Reviewer #1: All comments have been addressed

Reviewer #2: All comments have been addressed

2. Is the manuscript technically sound, and do the data support the conclusions?

Reviewer #1: Yes

Reviewer #2: Yes

3. Has the statistical analysis been performed appropriately and rigorously? 

Reviewer #1: Yes

Reviewer #2: Yes

4. Have the authors made all data underlying the findings in their manuscript fully available?

Reviewer #1: Yes

Reviewer #2: Yes

5. Is the manuscript presented in an intelligible fashion and written in standard English?

Reviewer #1: Yes

Reviewer #2: Yes

6. Review Comments to the Author

Reviewer #1: (No Response)

Reviewer #2: all the queries have been addressed and given references. congratulations for the effort and study undertaken.

7. PLOS authors have the option to publish the peer review history of their article (what does this mean?). If published, this will include your full peer review and any attached files.

Reviewer #1: No

Reviewer #2: No

---

## [Editor Report · Acceptance letter]

6 Oct 2022

PONE-D-22-20304R1 

Comparison of interproximal delivery and flow characteristics by dentifrice dilution and application of prepared toothpaste delivery technique 

Dear Dr. Satou:

I'm pleased to inform you that your manuscript has been deemed suitable for publication in PLOS ONE. Congratulations! Your manuscript is now with our production department. 

Kind regards, 

on behalf of

Dr. Tanay Chaubal 

Academic Editor

PLOS ONE